# Research on the Measurement of the Coordinated Relationship between Industrialization and Urbanization in the Inland Areas of Large Countries: A Case Study of Sichuan Province

**DOI:** 10.3390/ijerph192114301

**Published:** 2022-11-01

**Authors:** Lei Xiao, Jie Pan, Dongqi Sun, Zhipeng Zhang, Qian Zhao

**Affiliations:** 1China Academy of Urban Planning & Design Western Branch, Chongqing 401120, China; 2Institute of Geographic Sciences and Natural Resources Research, Chinese Academy of Sciences, Beijing 100101, China; 3Key Research Institute of Yellow River Civilization and Sustainable Development, Henan University, Kaifeng 475001, China

**Keywords:** industrialization, urbanization of large countries, coordinated relations, Sichuan province

## Abstract

Industrialization and urbanization are critical paths to modernization for a country or region. The coordination of industrialization and urbanization fosters the development of a regional economy. In academic circles, this is usually measured by the IU ratio (ratio of labor industrialization rate to urbanization rate) and the NU ratio (ratio of non-agricultural employment rate to urbanization rate). However, these methods are inapplicable to large countries’ inland areas. The traditional methods failed to explain the real situation and produced contradictory results. The IU ratio shows that industrialization lags behind urbanization, while the NU ratio shows that industrialization is ahead of urbanization. According to studies conducted in the Sichuan Province of China, through comparison with Jiangsu Province, it is found that the non-agricultural employment growth is not dependent on the development of local industrialization, and rural-urban migration is not entirely dependent on the evolution of the non-agricultural employment rate. Other factors that promote urbanization, such as the country’s capital policies and funds for migrant labor force transfer, should also be considered. This research attempts to improve the traditional methods for measuring the degree of urbanization and industrialization synergy in inland areas. The new empirical approach can effectively identify the critical characteristics of urbanization in inland provinces, such as the development of non-agricultural employment with external assistance and urban migrants “unrelated to employment opportunities”. Based on these key characteristics, it can provide the basis for local urbanization policy formulation.

## 1. Introduction

Industrialization is the development of industry on an extensive scale, and urbanization is the process of creating towns in country areas. The level of industrialization and urbanization is an essential standard for measuring regional development, and the coordination between them plays a critical role in regional sustainable development. Most studies on the relationship between urbanization, industrialization, and economic development are based on neoclassical growth theory, which holds that industrialization is the primary, urbanization is secondary, and industrialization is the foundation of urbanization and economic growth. Chenery and Serquin put forward the famous “pattern of development” theory by studying the economic development data of 101 countries worldwide [1]. They point out that the relationship between industrialization and urbanization is loosening processes. Furthermore, industrialization and urbanization are synchronous in developed countries, whereas in developing countries, urbanization lags behind industrialization. In China, industrialization is the primary driving force in promoting urbanization development. The coordination between urbanization and industrialization has attracted the attention of numerous scholars on the macro level, regional development level, and measurement methods.

On a broader level, compared with other countries, China’s urbanization development lags behind similar stages of industrialization development [2]. Therefore, most scholars believe that the relationship between industrialization and urbanization in China should be divided into two different stages: from the founding of the new China in 1949 to the initial stage of reform and opening-up and the adoption of “anti-urbanization” policies to promote industrial development has seriously hindered the coordinated development between urban and rural areas, resulting in urbanization lagging behind industrialization [3]. After the reform and opening-up, the new industrialization in China has effectively promoted the development of urbanization. Especially since the second half of the 1990s, the phenomenon of China’s urbanization lag has been gradually disappearing [4]. Cai [5] believed that although China’s urbanization process lags behind the industrialization process, it does not deviate too much from the industrialization process, and the level of industrialization and urbanization tends to be consistent. Wang [6] and Wei [7] pointed out that only when urbanization and industrialization are coordinated and complement each other, can we effectively ensure steady economic and social development. According to Song et al. [8], the rate of population agglomeration in China’s cities and towns is slower than the rate of industrial diffusion to rural areas, reflecting that China’s urbanization process lags behind industrialization. Duan and Zhang [9] discovered that China’s overall level of urbanization lags behind its industrialization. This lag phenomenon tends to diminish after the reform and opening-up. China’s urbanization and industrialization were most coordinated in 2003, but the disparity has gradually widened. According to Li and Wei [10], China’s coordinated development of industrialization and urbanization is increasing but remains low. Tertiary industry and urban infrastructure improvement are critical in promoting China’s coordinated development of industrialization and urbanization. From the dual perspective of urbanization in cities and towns, Zhang et al. [11] found that the coordination between urbanization and industrialization at the provincial level showed a trend of deterioration before improvement, and the time node of improvement is around 2005. Moreover, the positive interaction between urbanization in towns and industrialization continues to strengthen, while the competitive relationship with urbanization in cities is gradually emerging. Gu et al. [12] found that the migration of migrant workers in the new era is accompanied by six characteristics of industrialization and urbanization: low-cost industrialization; semi-urbanization; high-cost urbanization; dual characteristics of the process of citizenization; disorderly population flow; and the diversification and complexity of urban integration process. Besides, they proposed to promote the coordinated development of industrialization and urbanization by adjusting the flow direction and flow rate of migrant workers. Some scholars also believe that agriculture in China has developed many “blood transfusions” for industry and urban construction for a long time, resulting in insufficient “hematopoietic” function of agriculture itself. However, the level of Chinese industrialization and urbanization, as well as the comprehensive national strength, has now been improved. Therefore, China has basically had the economic strength of industry-feeding agriculture [13,14,15]. Some scholars believe specialization and agglomeration economy are the keys to coordinating industrialization and urbanization. The specialization economy caused by the division of labor promotes the deepening of industrialization and generates transaction costs, while the agglomeration economy can effectively reduce transaction costs and promote urbanization [16,17]. Ye and Huang [18] believe that the successive upgrading of industrial structure, and the large-scale and specialized development of enterprises, play a decisive role in the coordinated development of industrialization and urbanization.

At the regional development level, many scholars have conducted empirical analyses for urbanization and industrialization in the Beijing-Tianjin-Hebei, western, and inter-provincial level regions, respectively. Then, they both draw a similar conclusion that urbanization lags behind industrialization [19,20,21,22]. According to Duan and Zhang [9], urbanization in China’s eastern and central regions lags behind industrialization, while urbanization in the northeast region outpaces industrialization, and urbanization and industrialization in the western region are coordinated at a low level. Yuan [23] studied the new urbanization of Hubei Province. After using the entropy method, it was found that the development level of the whole province and city is very uneven. Wuhan City is far ahead, and the differences from other cities are obvious. Wang and Wang [24] believe that the coordination between urbanization and industrialization development in Sichuan Province is continuously rising. Li [25] revealed that Guizhou is uncoordinated, with industrialization lagging and urbanization outpacing. Xue et al. [26] found that there are some problems in the development of new-type urbanization in Hebei Province, and the two-level differentiation is relatively serious. They believed that it could be solved by changing the spatial layout of Hebei Province. Liang [27] found that the level of industrialization in Guangxi Province is low, and the level of urbanization is lower than the national average level, resulting in the level of industrialization lagging behind the level of urbanization, and the level of integrated development is gradually decreasing. Han [28] constructed a new urbanization index system from two dimensions, both inside and outside the province, and found that Sichuan Province has formed a highly unbalanced urban development pattern with Chengdu as the “dominant city” and other cities as the “stars holding the moon”. Wang et al. [29] discovered that Beijing’s urbanization has shifted from trailing industrialization in the 1990s to leading industrialization today. Ding [30] explored the relatively well-developed coastal areas, measured their coordination relationship by building a comprehensive evaluation index system, and found that they also have problems of poor coordination. Tu [31] conducted a study on the coordinated development of new industrialization and new urbanization in Hunan Province, and found that the regional disparity in the degree of coordination is obvious, showing the distribution of “high in the east, low in the west; high in urban agglomerations and low in the periphery”.

In terms of measurement methods, the ratio of labor industrialization rate to urbanization rate (IU ratio), and the ratio of labor non-agricultural employment rate to urbanization rate (NU ratio), are two typical methods for measuring the relationship between regional industrialization and urbanization development. Some domestic studies have studied and improved them in conjunction with China’s current situation. For example, Li [32] used the ratio between the change value of urban population proportion and the change value of agricultural labor force share over time as a proxy index to evaluate the coordinated development level of urbanization and industrialization in China, pointing out that the development of urbanization and industrialization is actually coordinated in most regions of China. Li and Wei [10] calculated the coordination degree by constructing two evaluation index systems of industrialization development level and urbanization development level and pointed out that the coordinated development degree of industrialization and urbanization in China showed an upward trend from 1978 to 2010, but remained at a low level overall. Liu [33] built a coordination degree model based on the comprehensive level index of industrialization and urbanization and analyzed the temporal and spatial distribution of the coordinated development of regional industrialization and urbanization in China since 1978. Wen [34] empirically analyzed the relationship between China’s new industrialization and new urbanization through the Granger causality test method. Based on the panel data of the Yangtze River Delta city group, Xie [35] constructed a new type of comprehensive evaluation index system for industrialization and industrialization, and used the coefficient of variation method to explore the degree of coordination. Xu et al. [36] studied 287 prefecture-level and higher cities in China in 2010. They measured the synchronous development level of urbanization, industrialization, informatization, and agricultural modernization (the new four modernizations), using the PLS path model and spatial distance measure model. According to their findings, China’s synchronous development of the new four modernizations presents the double contradiction of unbalanced development among regions and asynchronous development within areas. Liu and Zhao [37] analyzed the industrialization and urbanization development of Henan Province using three methods, including ADF, cointegration, and the Granger causality test. Deng and Zhang [38] studied the interactive effect of Sichuan’s industrialization and urbanization based on the VAR model after the reform and opening-up. Xiong et al. [39] used the panel data and PVAR model to explore the long-term dynamic equilibrium relationship between industrialization and urbanization by region. They found that industrialization has a significant boosting effect on urbanization, but the spatial carrying capacity of urbanization on industrialization is weak. Wang and Cao [40] selected 10 indicators of urbanization and industrialization to calculate the level score and the degree of coordination, and they found that the coordinated development of urbanization and industrialization tends to get better in general.

Current research frequently replaces the industrial output value with the secondary industry output value. The construction industry, therefore, has a significant impact on calculation data. In areas with backward economic development and relying on infrastructure construction, the proportion of construction industry is relatively prominent, and the calculation error of the correlation between urbanization and industrialization is more obvious, resulting in many conclusions inconsistent with the actual situation of the region. This paper examines the synergy of urbanization and industrialization in the inland areas of large countries. It uses Sichuan Province as an example to conduct empirical research to supplement existing research on coordinating urbanization and industrialization in China. Sichuan is a typical inland province, with a relatively weak industrial base compared with coastal provinces, and many labors have been lost, but in this case, it has still achieved rapid urbanization. The primary research ideas are as follows:Using the IU ratio and the NU ratio to measure the synergy between industrialization and urbanization in Sichuan Province, and comparing it with the situation in China and typical coastal provinces during the same period.Attempting to explain the interactive relationship between industrialization and urbanization in Sichuan Province, and determining the unique mechanism of agricultural population transfer in the inland areas of large countries.Attempting to develop a new measurement method capable of more accurately representing the coordination degree of industrialization and urbanization in such areas and including the local unique urbanization mechanism. To provide a new perspective and methodological support to study inland urbanization in large countries. Due to the epidemic’s impact after 2020, this paper mainly adopts the data before 2019.

## 2. Reflections of Traditional Methods

This study analyzes the application of traditional methods in some parts of large countries. It was found that the method of IU ratio and the NU ratio turned to different conclusions. The IU ratio shows that industrialization lags after urbanization, while the NU ratio shows that urbanization lags after industrialization. We proceed from the specific population structure to conduct an in-depth analysis of this difference.

### 2.1. Basic Information of Sichuan Province, China

This study uses Sichuan Province as the research object to investigate the coordination of urbanization and industrialization in the inland areas of large countries. China has the world’s largest population (as shown in Figure 1), with 1441.79 million people in 2019. Sichuan is the largest inland province in China, more than 2000 km away from the ocean, with a total area of 486,000 square kilometers and a population of 83.51 million in 2019. In the last 20 years, Sichuan has achieved remarkable urbanization development. Between 2000 to 2019, the urbanization rate increased rapidly from 26.7% to 55.4%, and the urban population increased from 21.99 million to 45.05 million, as shown in Table 1.

At the same time, Sichuan’s economic development level is relatively low in China. Based on the fixed price in 2000, Sichuan’s per capita GDP in 2019 was CNY 33,800, a significant increase from CNY 5000 in 2000. It is, however, lower than the national average level of CNY 37,100 in 2019 and significantly lower than Jiangsu (75,700), Guangdong (62,100), Zhejiang (67,000), and other coastal provinces.

### 2.2. The Contradiction between the Results of Empirical Research by Two Methods

To begin, this paper calculates the IU ratio (ratio of labor industrialization rate to urbanization rate) and the NU ratio (ratio of non-agricultural employment rate to urbanization rate) of Sichuan Province over the last 20 years. A high value indicates that industrialization has surpassed urbanization, while a low value suggests that industrialization has lagged. We also include the calculation results from Jiangsu Province in China for comparison, as shown in Figure 2 and Figure 3. Jiangsu Province is a coastal province in China with a high industrial accumulation and is widely regarded as a typical area where industrialization takes precedence over urbanization.

The results of the two methods appear to contradict one another. The IU ratio of Sichuan Province decreased from 0.39 to 0.28, which was always lower than the national average in the same year and significantly lower than that of Jiangsu Province. On the other hand, the NU ratio of Sichuan Province gradually decreased from 1.62 to 1.21, which was consistently higher than the national average level during the same period and significantly higher than the calculation results of Jiangsu Province. The NU ratio series shows that Sichuan’s industrialization is far ahead of urbanization, whereas the IU ratio series illustrates the opposite conclusion. The inconsistency between the two methods implies a distinct urbanization mechanism within a large country, making the traditional empirical research methods ineffective.

## 3. Analysis on the Causes of Contradictions

According to the above calculation results, the main reasons for the failure of traditional methods are found out by analyzing the specific process of urbanization and industrialization in Sichuan Province. We have divided the logic of “industrialization drives urbanization” into three links. First, there is industrial development, which absorbs agricultural labor into the industrial sector. Second, non-agricultural jobs grow as a result of industry. The third is the process by which agricultural labor force families relocate to cities and towns, leading to population migration from rural to urban areas. From this perspective, the study examines the unique mechanism in the urbanization process in Sichuan.

### 3.1. The Growth Rate of Industrial Employment Is Obviously Slower than That of Non-Agricultural Employment

From 2000 to 2019, the number of industrial employees in Sichuan increased by 2.54 million, from 4.83 million to 7.37 million. This figure appears to be significant, but it is not high in comparison to Sichuan’s total population of 83.51 million. The comparison between regions is more obvious. Although the population of Jiangsu Province is comparable to that of Sichuan, the industrial employment-population surpassed 20 million in 2010. Meanwhile, the urban population of Sichuan has increased by 23.06 million, which is nine times the increase in industrial employment during the same period.

The main reason is the massive outflow of the Sichuan laborers. From 2010 to 2020, more than 6 million laborers in Sichuan Province left to work in other provinces for an extended period, and only a few entered Sichuan from other places. Labor shortages slow the development of labor-intensive industrial sectors in Sichuan Province, such as textile, food, and other light industries. Sichuan’s industrial development has focused on the heavy industry sector, which has a limited capacity for job creation. Although industrial added value has increased significantly, employment growth has been relatively limited.

The overall growth momentum of non-agricultural jobs in Sichuan Province is relatively rapid. As shown in Figure 4, from 2000 to 2019, it increased by 11.58 million, from 20.15 million to 31.73 million. At the same time, the proportion of industrial employment in non-agricultural employment in Sichuan Province is only 23%, significantly lower than the 54% level in Jiangsu Province and the 35% national average level. The disparity reveals significant differences in the growth mechanism of non-agricultural employment between inland and coastal provinces in China, which is also the main reason for the discrepancy between the IU ratio and the NU ratio mentioned above.

### 3.2. The Capital Input Outside the Province Has Become an Important Reason for the Growth of Non-Agricultural Employment

External economic assistance has facilitated non-agricultural employment growth without requiring large-scale industrialization. As shown in Figure 5, compared with the traditional urbanization model driven by industrialization, Sichuan’s urbanization model is driven by external factors. First, the Chinese central government has invested heavily in “Western Development”, which has promoted large-scale infrastructure construction in inland areas. Second, there are 6 million migrant workers in Sichuan Province, and they spend a portion of their earnings on relatives back home, accounting for half of the local consumption capacity. Third, coastal provinces contribute more than half of Sichuan’s local fiscal expenditures. The funds from the three aspects have flowed into Sichuan’s construction, commercial, and public service industries, promoting rapid job growth in these sectors. As shown in Figure 6, in the structure of non-agricultural employment in Sichuan Province, the proportion of construction, consumer, and public service sectors are significantly higher. These sectors account for the vast majority of non-agricultural employment growth in Sichuan.

### 3.3. A Considerable Proportion of Urban Migrants Are “Irrelevant to Employment Opportunities”

According to the conventional view, new immigrants are primarily local non-agricultural employment practitioners and their relatives. However, a sample survey in Sichuan discovered that many immigrant families have no actual connection with local non-agricultural employment posts and belong to the migration to improve their lives [41]. According to the survey, many families in Sichuan Province have family members who work in other provinces of China. At the same time, the elderly and children remain in their hometowns and are commonly referred to as the “left-behind population”. They choose to relocate to cities and towns for education and pension, and the labor force working outside the province sends their earnings to them as a financial source for living in cities. The local employment statistics of Sichuan Province do not include these migrant workers, and this relocated “left-behind population” has become the province’s urban residents. This special mode can be summarized as remote industrialization driving local urbanization, as shown in Figure 7.

As a result, the urbanization process in local areas of large countries, in addition to being affected by the local economy, has economic and social links with external regions, which cannot be discussed separately in the study.

## 4. Establish New Methods

According to the above research, there is a unique urbanization mechanism in the inland areas of large countries, implying that the traditional development logic of “industrialization drives urbanization” is not fully established. Two factors must be considered. First, due to regional economic assistance, inland provinces can still grow non-agricultural employment in the absence of local industrialization. Second, due to labor outflow and the geographical separation of families, there are more urban migrants “unrelated to employment opportunities” in inland provinces.

### 4.1. Establish a New Index (PU Ratio) Suitable for the Inland Areas of Large Countries

The PU ratio is used as a new index to measure the coordination level of urbanization and industrialization. The specific calculation formula is:(1)$$\mathrm{PU}=\frac{\mathrm{ratio}\;\mathrm{of}\;\mathrm{the}\;\mathrm{production}\;\mathrm{sector}’s\;\mathrm{employment}\;\mathrm{to}\;\mathrm{the}\;\mathrm{total}\;\mathrm{employment}}{\mathrm{urbanization}\;\mathrm{rate}},$$

Among them,
the production sector’s employment = the number of industrial employments + “productive” service employment.(2)

The main change includes service departments, such as science and technology research and development, information services, business office, etc. The reason for this is that the economic growth environment has changed, and it may be biased to measure industrialization solely from the manufacturing perspective. As a result, some “productive” service sectors are included.

### 4.2. The Newly Established Derivative Index Represents the Critical Characteristics of Urbanization

The above PU ratio is decomposed into three derivative indexes to identify the primary causes of urbanization and industrial imbalance. The specific formula is as follows:(3)$$\begin{array}{ll} & \mathrm{PU}=\frac{\mathrm{ratio}\;\mathrm{of}\;\mathrm{the}\;\mathrm{production}\;\mathrm{sector}’s\;\mathrm{employment}\;\mathrm{to}\;\mathrm{the}\;\mathrm{total}\;\mathrm{employment}}{\mathrm{urbanization}\;\mathrm{rate}}\\ & =\frac{\mathrm{employment}\;\mathrm{in}\;\mathrm{production}\;\mathrm{sector}}{\;\mathrm{total}\;\mathrm{employment}}/\frac{\mathrm{urban}\;\mathrm{population}}{\mathrm{total}\;\mathrm{resident}\;\mathrm{population}}\\ & =\frac{\mathrm{employment}\;\mathrm{in}\;\mathrm{production}\;\mathrm{sector}}{\;\mathrm{non}-\mathrm{agricultural}\;\mathrm{employment}}\;*\;\frac{\mathrm{non}-\mathrm{agricultural}\;\mathrm{employment}\;}{\;\mathrm{urban}\;\mathrm{population}}/\frac{\mathrm{total}\;\mathrm{employment}}{\mathrm{total}\;\mathrm{resident}\;\mathrm{population}}\\ & =\frac{P\;*\;U}{E} \end{array}$$

P denotes the ratio of employment in the production sector to non-agricultural employment. The greater the *p*-value, the more significant the impact of industrialization on non-agricultural employment growth. If the *p*-value is low, it indicates that non-agricultural employment development has a minor relationship with local industrialization and is more influenced by external factors.

U denotes the ratio of non-agricultural employment to the urban population. The greater the value of U, the more farmers enter the city under employment guidance. Furthermore, the lower the U-value, the more life-oriented rural population migration into town. The more prominent the situation where the labor force works outside and the family members live in the area.

E denotes the total employment and resident population proportion, which serves as the control variable of the overall population structure. The higher the E-value, the greater the ratio of the employed population to the resident population and vice versa. The reason for setting this variable is that China’s internal labor force is pervasive across regions, and the birth rates vary significantly. Hence, determining the population structure is necessary.

## 5. Application of New Methods

The modified model is being applied to Sichuan Province and extended to all central provinces in China. The results show that the modified method is more in line with conventional cognition and can effectively distinguish the unique mechanism of urbanization reflected above. Therefore, it is vital to recognize the critical characteristics of urbanization in various regions and develop related policies.

### 5.1. PU Ratio Analysis: Sichuan’s Long-Term Industrialization Lags behind the Development of Urbanization

According to the calculation, in 2019, the PU ratio of Sichuan Province is 0.30, that of China is 0.39, and that of Jiangsu Province, a typical coastal province, is 0.57, as shown in Figure 8. Because most studies believe China’s overall industrialization and urbanization are coordinated, the national average PU ratio in 2019 serves as a reference threshold. The national PU ratio curve has been stable for a long time. It has slightly decreased, indicating that China has gradually progressed from relatively advanced industrialization to a coordinated level and has remained stable in recent years. Sichuan’s PU ratio curve has long been lower than the national average as a typical inland province. It has been in a relatively stable state for a long time, consistent with the high dependence of Sichuan’s cities and towns on state investment support after the western development. On the other hand, urbanization development is lagging in Jiangsu Province, and industrial workers are failing to integrate into urban life fully. The findings of the analysis are consistent with traditional cognition.

### 5.2. Comparison of Multiple Provinces: China’s Coastal and Inland Provinces Are Obviously Different

This method has been extended to other provinces in China. Given the comparability and availability of data, the comparison between areas is analyzed using the data from the sixth population census in 2010. According to the calculations shown in Table 2, China’s PU ratio in 2010 was 0.42, the average value of coastal provinces was 0.53, and the average value of inland provinces was 0.30. Urbanization has lagged after industrialization in most coastal areas. The provinces of Guangdong, Jiangsu, and Zhejiang, which are already “world factories”, have exceeded 0.6. Many industrial workers have not fully integrated into city life. However, urbanization has surpassed industrialization in most inland provinces. For example, the PU ratio of Gansu, Qinghai, Guizhou, Yunnan, Jilin, and Heilongjiang are lower than 0.3.

### 5.3. Derivative Index Analysis: Effectively Distinguish the Characteristics of Local Economic Structure and Population Structure

As shown in Figure 9, most provinces in China can be classified into the following types based on the levels of the P and U indexes:

The first category is “high-high” areas, also known as China’s coastal areas, which include Guangdong, Jiangsu, Zhejiang, Fujian, etc. Such areas have a strong economic foundation and a large inflow of population. The findings show that industrialization and non-agricultural employment are closely related (i.e., the P index is high), and the employment orientation of the urban population is clear (i.e., the U index is high).

The second category is “low-low” areas. In addition to Sichuan, it also includes Guizhou, Gansu, Qinghai, Shaanxi, and other underdeveloped western provinces, as well as northeast “rust areas” such as Liaoning, Jilin, and Heilongjiang. On the one hand, the ability of industrialization to drive the development of non-agricultural employment is limited, and the reliance on federal policy assistance is significant (i.e., the P index is low). On the other hand, there is a large outflow of the labor force, and more local urban migrants have nothing to do with employment opportunities (i.e., the U index is low).

The third category is “medium-low” areas, which include Shandong, Henan, Hubei, Hunan, Anhui, and Shanxi. The industrialization support capacity is close to the national average (i.e., the P index is not low), but there are significant labor outflows. Many immigration phenomena are “irrelevant to employment opportunities” (i.e., the U index is relatively low).

The situation in a few provinces is relatively special. For example, Hebei Province is close to Beijing and Tianjin. At the same time, it undertakes the main steel production in China. Therefore, its industrial employment is small, but its economy is large. The calculation results show that the P index is low and the U index is high. For another example, Yunnan Province has a weak industrialization foundation, but due to the prosperity of border trade, there is little population outflow. And the calculation results of Yunnan Province show that the P index is low and the U index is relatively middle.

## 6. Discussing and Summary

### 6.1. Discussing: Urbanization Policies of Large Countries Need to Be Based on Regional Differences

Most studies on urbanization in China focus on coastal provinces, with a strong industrialization foundation and a high labor inflow. In this context, many studies have taken “how to integrate migrant farmers into the city” as the core issue of China’s urbanization, which has also become an essential focus of China’s urbanization policymaking. However, China’s industrialization progress has been uneven. Most inland provinces have a low level of industrialization and a high labor outflow. Their problems are not the same as those described above. Therefore, the critical starting point of this paper is to incorporate the case of China’s inland areas into a unified analysis framework and to develop differentiated regional policies.

The improved method can be used to develop urbanization policies in some areas of China. If a region has a high PU ratio, its industrialization is advanced, but urbanization lags. If the PU ratio is low, the region’s urbanization is ahead of schedule, but its industrialization is insufficient. China’s provinces can be divided into two policy areas that implement different policies to coordinate industrialization and urbanization. The policy priorities in specific regions can be refined according to the calculation results of the P and U indexes.

The P index is used to determine whether there are deficiencies in the development of local industrialization and can serve as the basis for regional economic policymaking. Areas with a high P index should encourage the development of urban commerce and public service departments. Conversely, if the P index is low, it is necessary to strengthen the local industrial-economic support, actively expand the employment absorption capacity of the industrial sector, and avoid the local economy’s excessive reliance on real estate and infrastructure investment.

The U index measures the proportion of employed and non-employed people in the urban population. It can be used to guide the allocation of various urban facilities. China’s urban planning typically follows similar construction standards across cities but may deviate from reality in different regions. Areas with a high U index should pay more attention to the demand for production facilities and lower the threshold for immigrants to live in the local area. Moreover, areas with a low U index should devote more policy resources to the living security of urban migrants, and the proportion of public facilities in the unit population should be increased, such as education, medical treatment, and parks, etc.

### 6.2. Main Conclusion: Improvement of a Measurement Method for the Urbanization and Industrialization in Inland Areas of Large Countries

Using the Sichuan Province of China as the research area, this paper discusses the synergy between urbanization and industrialization in the inland regions of large countries. In the empirical analysis of Sichuan, it was discovered that the traditional IU ratio and the NU ratio methods yield contradictory results, and the urbanization of Sichuan is not a simple conventional cognition of “industrialization drives urbanization”.

Further studies have revealed two critical aspects of the urbanization process in Sichuan. First, inland provinces of large countries frequently receive economic assistance from the central government and other provinces. In addition to local industrialization, they can gain additional impetus to develop non-agricultural employment, as evidenced by the rapid expansion of employment sectors such as construction and living consumption. Second, most inland provinces of large countries have a labor force working in other provinces, while many family members of the migrant labor force choose to live in the region. As a result, statistically, some more urban immigrants have nothing to do with employment opportunities, such as the elderly entering the city for better living conditions and children for better education.

The findings are incorporated into the urbanization analysis framework, and traditional measurement methods are modified. Using the revised PU ratio to measure the correlation degree between urbanization and industrialization within large countries, it was discovered that China’s coastal areas are essentially ahead of industrialization, while the inland regions are ahead of urbanization. We can now distinguish the characteristics of urbanization in China using the new P and U indexes. The lower the P index, the greater the gap between industrialization and urbanization, and the more obvious the reliance of urbanization on external assistance. Places with a low U index tend to have severe labor outflow and a greater number of immigrants “unrelated to employment opportunities” in the process of urbanization. The calculated results are consistent with the conventional cognition results.

## Figures and Tables

**Figure 1 ijerph-19-14301-f001:**
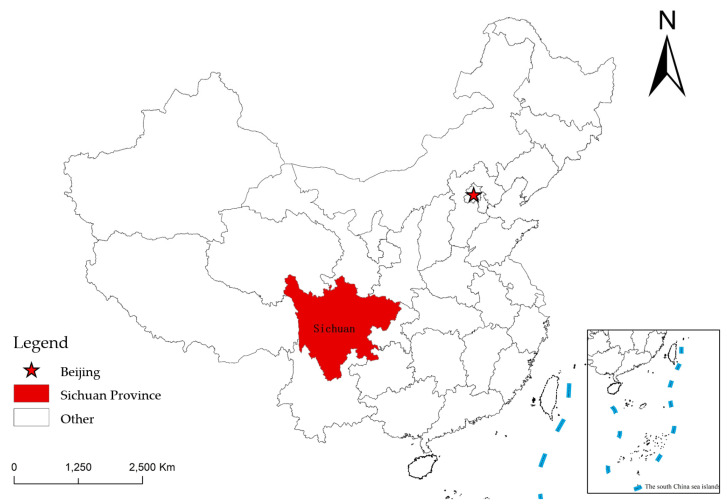
Regional map of Sichuan Province in China (created by ArcGIS).

**Figure 2 ijerph-19-14301-f002:**
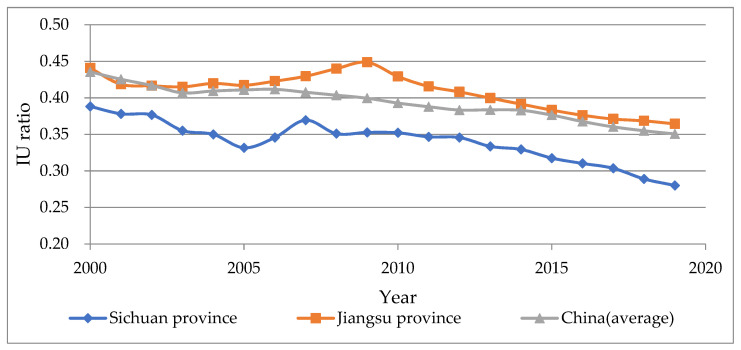
The calculation results and comparative analysis of IU ratio in Sichuan Province from 2000 to 2019 (data sources are *Sichuan Statistical Yearbook 2020*, *Jiangsu Statistical Yearbook 2020*, and *China Statistical Yearbook 2020*).

**Figure 3 ijerph-19-14301-f003:**
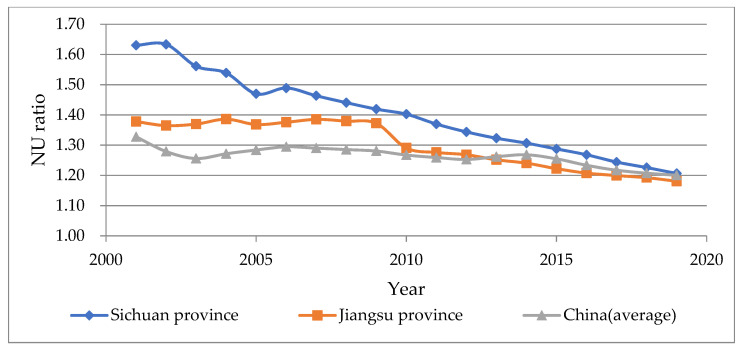
The calculation results and comparative analysis of NU ratio in Sichuan Province from 2001 to 2019 (data sources are *Sichuan Statistical Yearbook 2020*, *Jiangsu Statistical Yearbook 2020*, and *China Statistical Yearbook 2020*).

**Figure 4 ijerph-19-14301-f004:**
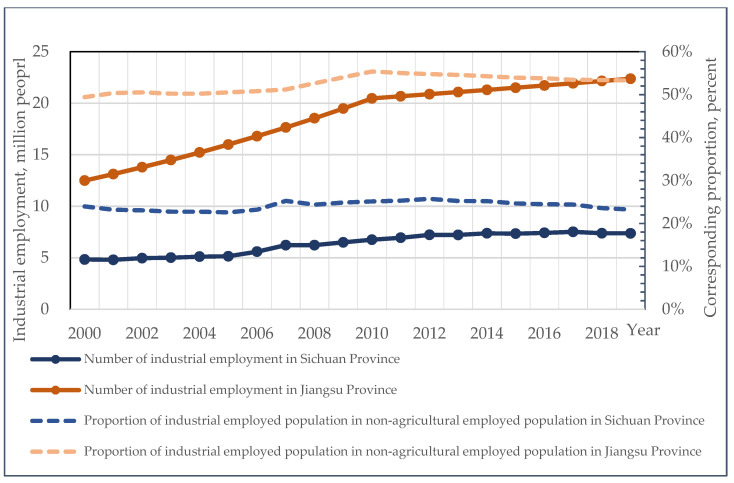
Comparison of industrial employment in Sichuan and Jiangsu from 2000 to 2019 (data sources are *Sichuan Statistical Yearbook 2020* and *Jiangsu Statistical Yearbook 2020*).

**Figure 5 ijerph-19-14301-f005:**
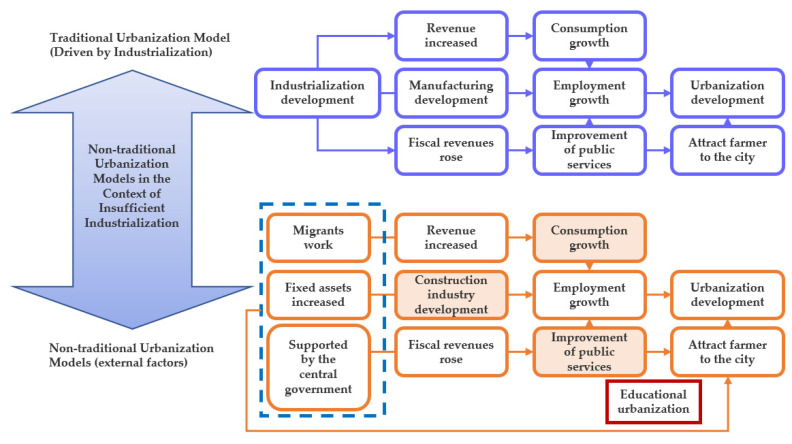
The schematic diagram of traditional and non-traditional urbanization mechanism.

**Figure 6 ijerph-19-14301-f006:**
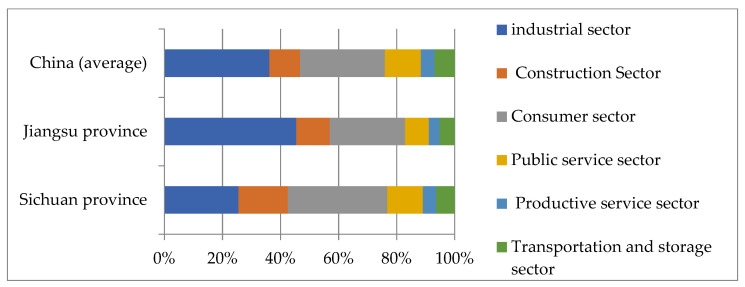
Comparative analysis of sub-sector structure of non-agricultural employment in Sichuan (data sources are *Sichuan Statistical Yearbook 2020*, *Jiangsu Statistical Yearbook 2020*, and *China Statistical Yearbook 2020*).

**Figure 7 ijerph-19-14301-f007:**
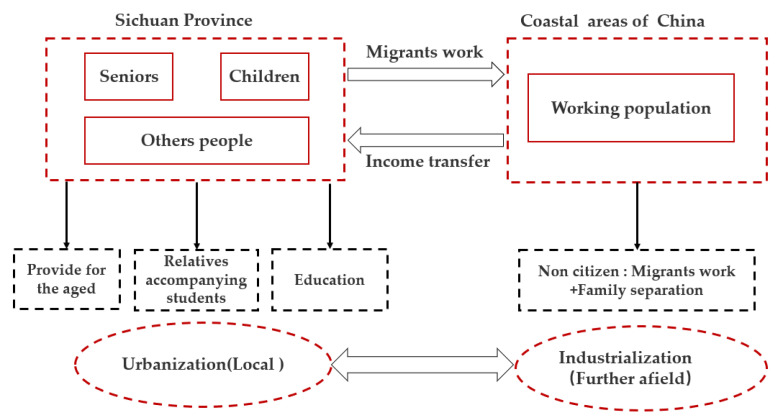
Schematic diagram of the mode in which migrant workers in Sichuan drive the local population to migrate to cities and towns.

**Figure 8 ijerph-19-14301-f008:**
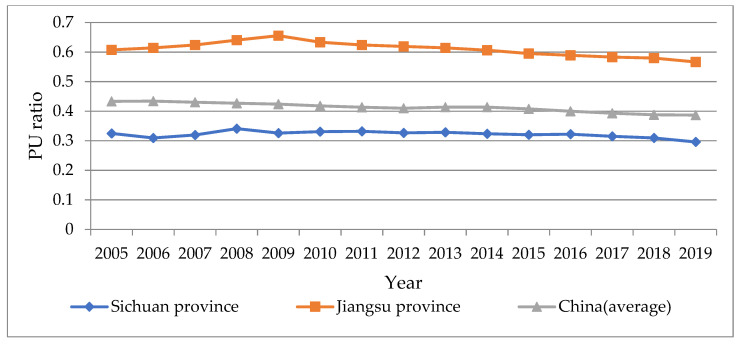
The calculation results and comparative analysis of PU ratio from 2005 to 2019 (data sources are *Sichuan Statistical Yearbook 2020*, *Jiangsu Statistical Yearbook 2020*, and *China Statistical Yearbook 2020*).

**Figure 9 ijerph-19-14301-f009:**
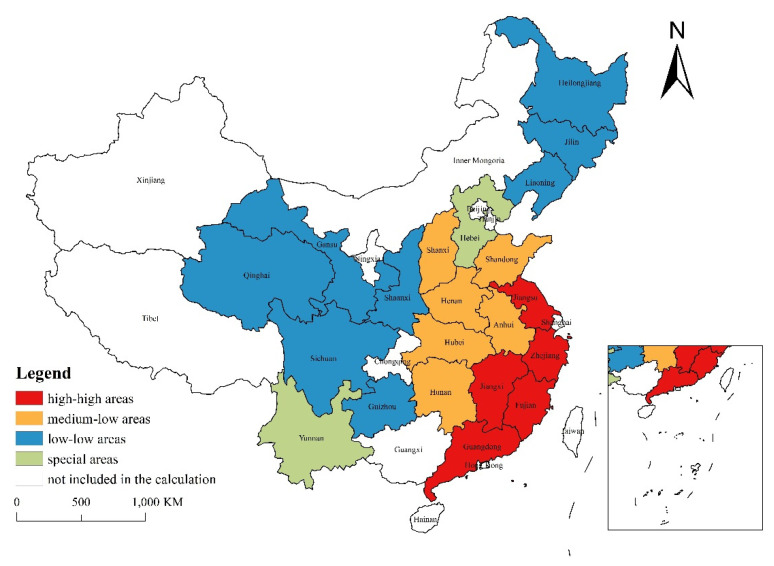
Schematic diagram of classification of some Chinese provinces based on P and U values in 2010 (created by ArcGIS, and data source is the sixth population census of China).

**Table 1 ijerph-19-14301-t001:** Primary data of population urbanization in Sichuan Province of China from 2000 to 2019.

Year	Permanent Resident Population (Million)	Urbanization Rate (%)	Non-Agricultural Employment (Million)	Number of Industrial Employees (Million)
2000	82.35	26.7	20.15	4.83
2001	81.43	27.2	20.70	4.80
2002	81.10	28.2	21.50	4.96
2003	81.76	30.1	22.01	5.01
2004	80.90	31.1	22.45	5.11
2005	82.12	33.0	22.81	5.15
2006	81.69	34.3	24.08	5.59
2007	81.27	35.6	24.65	6.22
2008	81.38	37.4	25.54	6.22
2009	81.85	38.7	26.12	6.49
2010	80.45	40.2	26.89	6.75
2011	80.64	41.8	27.42	6.94
2012	80.85	43.4	28.07	7.22
2013	81.09	45.0	28.62	7.22
2014	81.39	46.5	29.24	7.37
2015	81.96	48.3	29.76	7.34
2016	82.51	50.0	30.33	7.42
2017	82.89	51.8	30.79	7.51
2018	83.21	53.5	31.29	7.38
2019	83.51	55.4	31.73	7.38

Data source: Sichuan Statistical Yearbook 2020.

**Table 2 ijerph-19-14301-t002:** The calculation results of PU ratio and related indexes in central provinces of China in 2010.

Area	PU Ratio	P Index (Industry Supporting Capacity)	U Index (Employment Oriented Urbanization)	E Index (Employment Proportion of Population)	Per Capita GDP (yuan)
National average	0.42	0.40	0.56	0.54	29,668
Jiangsu	0.63	0.49	0.73	0.57	52,840
Zhejiang	0.78	0.56	0.83	0.60	51,711
Fujian	0.58	0.46	0.66	0.53	40,025
Shandong	0.40	0.44	0.54	0.59	41,106
Guangdong	0.65	0.57	0.60	0.53	44,736
Jiangxi	0.56	0.44	0.64	0.51	21,253
Guizhou	0.26	0.29	0.44	0.48	13,119
Shaanxi	0.28	0.28	0.53	0.53	27,133
Gansu	0.21	0.26	0.44	0.55	16,113
Qinghai	0.23	0.24	0.49	0.52	24,115
Jilin	0.21	0.28	0.40	0.52	31,599
Heilongjiang	0.24	0.30	0.39	0.49	27,076
Liaoning	0.32	0.36	0.48	0.54	42,355
Hebei	0.37	0.33	0.63	0.56	28,668
Shanxi	0.37	0.35	0.51	0.48	26,283
Anhui	0.36	0.33	0.52	0.49	20,888
Henan	0.30	0.34	0.46	0.53	24,446
Hubei	0.33	0.35	0.50	0.53	27,906
Hunan	0.37	0.36	0.53	0.52	24,719
Yunnan	0.26	0.29	0.51	0.58	15,752

Note: Due to the difference in economic structure, the municipalities directly under the central government, autonomous regions, particular administrative regions, and Hainan Province with restrictions on the industry are not listed here. Additionally, due to the different statistical caliber of the census data of Taiwan, Hong Kong and Macao, these three places are not included here.

## Data Availability

Not applicable.

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
