# Peer review of "Research on the Measurement of the Coordinated Relationship between Industrialization and Urbanization in the Inland Areas of Large Countries: A Case Study of Sichuan Province"

_ijerph, 2022, doi:10.3390/ijerph192114301_

Round 1

Reviewer 1 Report

Dear Authors,

It is interesting that urbanization lags behind industrialization in China [19-22].

Also, it is interesting that “Current research frequently replaces the industrial output value with the secondary industry output value. The construction industry, therefore, has a significant impact on calculation data.”.

Lines 263-266

Why differences in the growth mechanism of non-agricultural employee in Sichuan was occurred? 

Authors analyzed a detailed parameter for differences in the growth mechanism of non-agricultural employee in Sichuan. If we use your analysis equations, how many years will need to get more accurate data?

Sincerely,

Author Response

Dear Reviewer,

We are very grateful to your comments for the manuscript. Some of your questions were answered below.

Point 1: Lines 263-266

Why differences in the growth mechanism of non-agricultural employee in Sichuan was occurred?

Response 1: Specific explanations are covered in Section 3.2. In the traditional industrialization theory, a certain amount of manufacturing employment drives other corresponding non-agricultural employment. But due to the inflow of external funds, non-agricultural employee in Sichuan has gained additional development when there is not much manufacturing. The main source of funds is the income of migrant workers. Migrant workers send their income back to their parents to take care of their children in their hometown.

Point 2: Authors analyzed a detailed parameter for differences in the growth mechanism of non-agricultural employee in Sichuan. If we use your analysis equations, how many years will need to get more accurate data?

Response 2: If you want to apply the new method, you only need data of a single year, from which you can judge the sequence of industrialization and urbanization, as well as the impact of population structure and external capital input. China carried out the seventh population census in 2020, but detailed data have not been released. At present, the most accurate data available is still the 2010 census data.

Reviewer 2 Report

1. A map should be drawn as a supplement to Table 2 since you conclude that the lag between industrialization and urbanization shall be analyzed at a regional level. The four categories in Section 5.3 might also be distinguished by different colors on the map.

2. On page 6, the author describes Sichuan Province's IU and NU results. However, the discussion of Jiangsu Province is missing although the result is normal.  

3. The formulas in Section 4.1 are written irregularly. The superscript and subscript could help with a better-written style.

And some advice on formatting.

4. All the figures and tables are not cited in the text.

5. The number (in section 2.1) shall use commas for a better reading experience, i.e., 486000 km2 shall be written as 486,000 km2;

6. The name shall be the Nine-dotted Line instead of the nine-dashed line legend in Figure 1.

7. There are some synonyms with a slash in the text, I think the author shall check carefully and keep one.  For example, obvious/pronounced, Simultaneously/Meanwhile in Section 3.1.

Author Response

Dear Reviewer,

We are very grateful to your comments for the manuscript. According with your advice, we amended the relevant part in manuscript. Some of your questions were answered below.

Point 1: A map should be drawn as a supplement to Table 2 since you conclude that the lag between industrialization and urbanization shall be analyzed at a regional level. The four categories in Section 5.3 might also be distinguished by different colors on the map.

Response 1: Figure 9 is added as a supplement to Table 2.

Point 2: On page 6, the author describes Sichuan Province's IU and NU results. However, the discussion of Jiangsu Province is missing although the result is normal.

Response 2: Jiangsu Province has always been regarded as a place with leading industrialization and urbanization. From the calculation results, it is the same. There is no obvious contradiction, so they didn't expand in detail. The situation in Jiangsu Province is different from that in Sichuan Province. Jiangsu Province is not an area where the income of migrant workers is imported. On the contrary, there are many migrant workers who send their income back to their hometown instead of sending their families to the cities where they work.

Point 3: The formulas in Section 4.1 are written irregularly. The superscript and subscript could help with a better-written style.

Response 3: The formulas in Section 4.1 and 4.2 have been modified to use superscript and subscript.

Point 4: All the figures and tables are not cited in the text.

Response 4: All the figures and tables are cited after modification.

Point 5: The number (in section 2.1) shall use commas for a better reading experience, i.e., 486000 km2 shall be written as 486,000 km2.

Response 5: The number (in section 2.1) has been modified to use commas.

Point 6: The name shall be the Nine-dotted Line instead of the nine-dashed line legend in Figure 1.

Response 6: Legend in Figure 1 has been modified.

Point 7: There are some synonyms with a slash in the text, I think the author shall check carefully and keep one.  For example, obvious/pronounced, Simultaneously/Meanwhile in Section 3.1.

Response 7: We carefully proof-read the manuscript again to minimize typographical,grammatical,and bibliographical errors, and only keep one synonym.

Reviewer 3 Report

In this work, the authors provide a new empirical methodology to identify the synergetic effect of industrialization and urbanization in inland areas, resulting in consolidated results compared to traditional methods. The work is very focused and well-written. The review component of the introduction is detailed and thorough. The analysis of the results is systematic and assists the reader in reaching the same conclusions. However, the methodology on the traditional methods, and source of data are missing in this work. The quality of the figures has to be enhanced, and in-text mention should be added. The work can certainly be published in this journal, however, with some key comments to be addressed:

1.     Line 15, since this is mentioned here for the first time, the acronyms IU and UN ratio need to be defined.

2.     Line 17, a brief description of the contradiction in the results from traditional methods need to be mentioned.

3.     Line 26, a small capping phrase is needed, for the direction of research or progress to follow relying on the new empirical approach developed in this work.

4.     Within the first opening lines in the introduction, a small distinction between industrialization and urbanization is needed for the non-experts in the area.

5.     Line 36, the ref number should be added after mentioning the work from other authors.

6.     Line 168, the rational behind choosing Sichuan Province to conduct the analysis is needed, should be short and focused as additional details are added in subsequent sections.

7.     Lines 183-186 are filled with contentious statements, as the authors claim issues with the traditional methods, prior to showing the results. This should be rephrased.  

8.     Table 1 is not mentioned in text, also, the source of these data should be added.

9.     The authors should include a small section defining the IU and NU ratio and their computational requirements as they are missing from the manuscript.

10.  Figures 2 and 3 are missing the title for the y-axis and x-axis.

11.  In section 2.2, in-text mention of figures 2 and 3 are missing.

12.  In Section 3.1, in-text mention of figure 4 is missing.

13.  Figure 4, the primary and secondary y-axis titles are missing.

14.  Source of data used to build figure 5 are missing, the same goes for the data in figure 4.

15.  Figure 6 is misplaced and does not follow with the narrative of the results that the authors are presenting. Also, the analysis of the flow included in the figure is missing from the description of the results.

16.  Again, figure 7 is not mentioned in the text and not included in the discussion of the results.

17.  Figure 8 is missing its y-axis title, and not mentioned in the text.

18.  At the end of the conclusions, a short statement on the added value of this new analysis metric is needed, and how it can be effectively used in future policy setting frameworks.

Author Response

Dear Reviewer,

We are very grateful to your comments for the manuscript. According with your advice, we amended the relevant part in manuscript. Some of your questions were answered below.

Point 1: Line 15, since this is mentioned here for the first time, the acronyms IU and UN ratio need to be defined.

Response 1: We added an explanation of IU and NU ratio. IU ratio is the ratio of labor industrialization rate to urbanization rate, and NU ratio is the ratio of non-agricultural employment rate to urbanization rate.

Point 2: Line 17, a brief description of the contradiction in the results from traditional methods need to be mentioned.

Response 2: A brief description of the contradiction has been added. IU ratio shows that industrialization lags behind urbanization, while NU ratio shows that industrialization is ahead of urbanization.

Point 3: Line 26, a small capping phrase is needed, for the direction of research or progress to follow relying on the new empirical approach developed in this work.

Response 3: The expression in the text has been modified as required.

Point 4: Within the first opening lines in the introduction, a small distinction between industrialization and urbanization is needed for the non-experts in the area.

Response 4: The explanation of industrialization and urbanization has been added at the begginning of the paragraph.

Point 5: Line 36, the ref number should be added after mentioning the work from other authors.

Response 5: The reference number is adjusted to the proper position.

Point 6: Line 168, the rational behind choosing Sichuan Province to conduct the analysis is needed, should be short and focused as additional details are added in subsequent sections.

Response 6: The reason for choosing Sichuan Province is added after the sentence. “Sichuan is a typical inland province, with a relatively weak industrial base compared with coastal provinces, and many labors have gone out, but in this case, it has still achieved rapid urbanization.”

Point 7: Lines 183-186 are filled with contentious statements, as the authors claim issues with the traditional methods, prior to showing the results. This should be rephrased.

Response 7: The whole paragraph was rewritten to make the expression more accurate.

Point 8: Table 1 is not mentioned in text, also, the source of these data should be added.

Response 8: After modification, Table 1 is mentioned in the corresponding position of the text, and data sources are also added.

Point 9: The authors should include a small section defining the IU and NU ratio and their computational requirements as they are missing from the manuscript.

Response 9: The definition and calculation requirements of IU and NU ratio are mentioned in Section 2.2.

Point 10: Figures 2 and 3 are missing the title for the y-axis and x-axis.

Response 10: The titles for the y-axis and x-axis of Figures 2 and 3 have been added.

Point 11: In section 2.2, in-text mention of figures 2 and 3 are missing.

Response 11: After modification, Figure 2 and 3 are mentioned in the corresponding position in section 2.2.

Point 12: In Section 3.1, in-text mention of figure 4 is missing.

Response 12: After modification, Figure 4 is mentioned in the corresponding position in section 3.1.

Point 13: Figure 4, the primary and secondary y-axis titles are missing.

Response 13: The primary and secondary y-axis titles of Figures 4 have been added.

Point 14: Source of data used to build figure 5 are missing, the same goes for the data in figure 4.

Response 14: After modification, data sources of Figure 4 and 5 have been added.

Point 15: Figure 6 is misplaced and does not follow with the narrative of the results that the authors are presenting. Also, the analysis of the flow included in the figure is missing from the description of the results.

Response 15: Figure 5 and Figure 6 switch places to ensure that they are in the right place. The corresponding narrative has also been added to the text.

Point 16: Again, figure 7 is not mentioned in the text and not included in the discussion of the results..

Response 16: After modification, Figure 7 is mentioned in the corresponding position in section 3.3.

Point 17: Figure 8 is missing its y-axis title, and not mentioned in the text.

Response 17: After modification, Figure 8 is mentioned in the corresponding position in section 5.1. Also its titles for the y-axis and x-axis and data sources have been added.

Point 18: At the end of the conclusions, a short statement on the added value of this new analysis metric is needed, and how it can be effectively used in future policy setting frameworks.

Response 18: The significance and usage of the new analysis metric are explained in Section 6.1 Discussing. The P index is used to measure whether the local economic structure is reasonable. The regions with high p index indicate that the life type sector in the economic structure is relatively small, and the development of commerce and public services needs to be increased. The regions with low P index indicate the lack of industrialization development and need to expand manufacturing and other productive sectors to avoid excessive dependence of local economy on real estate and infrastructure investment. The U index is used to measure the proportion of employed people and non employed people in the urban population. In areas with high U index, it shows that there are more work oriented people among migrants. In the construction of urban facilities, more attention should be paid to the facilities needed for production, and more support should be given to rural migration policies, so that migrants are more willing to send their families to cities where they work. In areas with low U index, there are fewer migrants working and more elderly and children. In this case, local governments need to increase more policy resources for the livelihood security of migrants and increase the proportion of public facilities in the unit population, such as education, medical care, parks, etc.

Reviewer 4 Report

The manuscript is an interesting case study on the development of urbanization in the Sichuan Province of China. The data show an interesting relationship between population growth at the level of urbanization of selected province in last 20 years.
In the paper also interesting are the relationships between the two discussed regions in China with the original data for China, as shown in clear graphs. The statistics in the given paper are well presented, substantiated and discussed.
The paper clearly illustrates the stages of development of two types of urbanization, traditional and non-traditional, which is well illustrated in a clear framework. The paper has an original and innovative character. The main thread of the article is the proposal of the proprietary PU-Ratio method, which was presented in diagrams against the background of the provinces discussed and the whole country. The method can be applied in similar case studies for large provinces in large countries such as China. It makes sense that similar studies can be continued or updated in the future and compared with other provinces in a country in China (like the same as it is now interesting compared with Jangsu province). Certainly, the indicator introduced could be used more widely. However, this should be the subject of further investigation, in a separate paper as continuation of the given research.

The only three things I would suggest are:
- adding the source and the software used to create the maps,
-adding the information of Jangsu province in the abstract or in the title, cause some data are also presented on it in the paper,
- rearranging the Chapter 6 so that the discussion takes precedence over the summary. However, I leave these three remarks to the decision of the authors.
I think the work meets the editorial requirements and is suitable for publication in the JERPH Journal.

Author Response

Dear reviewer:

We are very grateful to your comments for the manuscript. According with your advice, we amended the relevant part in manuscript. Some of your questions were answered below.

Point 1: adding the source and the software used to create the maps.

Response 1: The source and the software used to create the maps have been added to the corresponding figure title.

Point 2: adding the information of Jangsu province in the abstract or in the title, cause some data are also presented on it in the paper.

Response 2: The information of Jangsu province has been added in the abstract.

Point 3: rearranging the Chapter 6 so that the discussion takes precedence over the summary.

Response 3: Section 6.1 conclusion and section 6.2 Discussing switch places to ensure that they are in the right place.
